# Fish Consumption and DHA Supplementation during Pregnancy: Study of Gestational and Neonatal Outcomes

**DOI:** 10.3390/nu16183051

**Published:** 2024-09-10

**Authors:** Paola Gualtieri, Giulia Frank, Rossella Cianci, Francesca Dominici, Ilenia Mappa, Giuseppe Rizzo, Gemma Lou De Santis, Giulia Bigioni, Laura Di Renzo

**Affiliations:** 1Section of Clinical Nutrition and Nutrigenomics, Department of Biomedicine and Prevention, University of Tor Vergata, Via Montpellier 1, 00133 Rome, Italygemmaloudesantis@gmail.com (G.L.D.S.); bigionigiulia@gmail.com (G.B.); laura.di.renzo@uniroma2.it (L.D.R.); 2PhD School of Applied Medical-Surgical Sciences, University of Tor Vergata, Via Montpellier 1, 00133 Rome, Italy; giulia.frank@ymail.com; 3School of Specialization in Food Science, University of Tor Vergata, Via Montpellier 1, 00133 Rome, Italy; francescadominici1991@gmail.com; 4Department of Translational Medicine and Surgery, Catholic University of the Sacred Heart, 00168 Rome, Italy; 5Fondazione Policlinico Universitario A. Gemelli, Istituto di Ricovero e Cura a Carattere Scientifico (IRCCS), 00168 Rome, Italy; 6Department of Maternal Fetal Medicine, Ospedale Cristo Re, 00167 Rome, Italy; mappa.ile@gmail.com; 7Department of Maternal and Child Health and Urological Sciences, Sapienza University of Rome, Policlinico Umberto I, Viale Regina Elena 324, 00161 Rome, Italy; giuseppe.rizzo@uniroma1.it

**Keywords:** pregnancy, docosahexaenoic acid, supplementation, fish, consumption

## Abstract

Several studies have explored the association between fish consumption during pregnancy and favorable neonatal outcomes, although some yield conflicting results. The American College of Obstetricians and Gynecologists recommends two to three servings of low-mercury fish per week for pregnant or breastfeeding women. However, fish can be a source of pollutants, like methylmercury, impacting neurological development. Conflicting studies on docosahexaenoic acid (DHA) supplementation exist in the literature, possibly due to varied supplement dosages. This survey, involving 501 women, investigated fish consumption and DHA supplement intake concerning gestational and neonatal outcomes. Notably, 92.1% of participants consumed fish weekly, with significant differences observed in gestational weight gain, birth weight, and length for those eating fish ≥3 times weekly compared to non-consumers. This study supports the recommendation for pregnant women to include fish in their diet while limiting exposure to environmental pollutants. Omega-3 fatty acid supplements are suggested to attain nutritional benefits without mercury risk.

## 1. Introduction

An adequate maternal nutritional state, even before the onset of pregnancy, along with proper nutrition throughout pregnancy, is essential for preventing numerous maternal–fetal pathologies. Fish is nutritionally valuable due to its high-quality protein, long-chain polyunsaturated omega-3 fatty acids (LCPUFA *n*-3), vitamin D, iodine, and selenium, all deemed advantageous for fetal growth and development. While many studies show favorable neonatal outcomes associated with fish consumption during pregnancy [1], some present conflicting results [2,3]. Worldwide, there has been an increase in per capita fish consumption from nearly 10 kg in 1960 to over 20 kg in 2014 [4]. Italy, a major European market, recorded a per capita consumption of 30.9 kg in 2017 [5]. Recommendations by the American Heart Association [6] and the American College of Obstetricians and Gynecologists [7] suggest two servings/per week (pw) of low-mercury fish for pregnant or breastfeeding women. Nevertheless, fish can serve as a pathway for exposure to pollutants, like methylmercury, polychlorinated biphenyls, dioxins, and other heavy metals, which may have adverse effects [8].

The Environmental Protection Agency (EPA) and the Food and Drug Administration (FDA) advise pregnant women to avoid high-mercury fish species (shark, swordfish, and king mackerel) and favor lower-mercury options (salmon, shrimp, canned fish, catfish, and cod) [9]. These recommendations aim to provide sufficient docosahexaenoic acid (DHA) to the developing fetus while safeguarding the maturing nervous system from the neurotoxic effects of mercury. Containing both favorable and unfavorable substances for fetal health and pregnancy progression, the overall effect of fish consumption during pregnancy remains uncertain.

Observational studies and clinical trials have demonstrated associations between fish consumption and reduced preterm birth rates, prolonged gestation, and increased birth weight [10,11,12,13,14,15,16]. However, conflicting results exist [2,17,18,19,20,21,22], possibly due to variations in nutrient content among fish types and patient consumption habits. Fatty fish contains more *n*-3, while shellfish may contain more pollutants [23,24]. Similarly, conflict in the literature exists regarding DHA supplementation, likely due to varying supplement dosages. Pregnancy is linked to a decrease in maternal serum DHA levels and the potential depletion of maternal reserves. [25]. Hence, the DHA requirement increases by 100–200 mg/day during pregnancy [26]. Only 2% of pregnant women have a diet meeting the correct DHA requirement, leading to an imbalance in the *n*-6/*n*-3 ratio, particularly in the Mediterranean diet [27].

The Italian Society of Human Nutrition recommends an *n*-6/*n*-3 ratio of at least 5:1 for pregnant women, while epidemiological data indicate that 85% of women of childbearing age are deficient in DHA [28]. This deficiency prompts the recommendation for DHA supplementation even before pregnancy [29].

Using a survey, our study aims to investigate fish and DHA supplement intake concerning gestational and neonatal outcomes. For gestational outcomes, we explore whether fish and/or DHA consumption correlates with favorable weight gain during pregnancy, the type of delivery (natural or cesarean), and the gestational week of delivery. For neonatal outcomes, we examine birth weight and length. As secondary analyses, we explore associations between fish consumption or DHA supplementation and preterm birth, as well as the interaction between pre-pregnancy BMI and fish consumption or DHA intake on neonatal outcomes.

## 2. Materials and Methods

### 2.1. Study Design

This observational study was conducted by the Clinical Nutrition and Nutrigenomics Section, Department of Biomedicine and Prevention at the University of Rome Tor Vergata. A web survey was employed to gather data on dietary habits and lifestyle during pregnancy. The survey was conducted from November 2022 to July 2023, targeting Italian patients who gave birth during that period, belonging to the Division of Gynecology and Obstetrics of the Department of Biomedicine and Prevention at Tor Vergata University, and to the Department of Maternal Fetal Medicine of Cristo Re Hospital. Distribution was carried out through an online platform accessible from any internet-connected device, and utilized institutional and private social networks (Twitter, Facebook, and Instagram), the “PATTO in Cucina Magazine” website, and institutional mailing lists, with a snowball method. Exclusions comprised questionnaires from vegetarian, vegan, and twin pregnancy participants. The questionnaire was structured into 36 questions divided into 5 sections: (a) personal data (3 questions); (b) anthropometric data (3 questions); (c) pregnancy session (14 questions); (d) dietary and lifestyle habits (10 questions); (e) eating habits, focused on DHA and fish (6 questions). Regarding the doses of DHA, no questions were asked, as the supplements available in Italy have a dose of 200 mg. The questionnaire is available in Appendix A. This research adhered to national and international regulations and the Declaration of Helsinki (2000). Participants were fully informed of study requirements and consented to data sharing and privacy policies before participation. The Google-linked questionnaire ensured participant anonymity, with personal information, including names, anonymized for confidentiality. This study was approved by the Ethics Committee of the Calabria Region Center Area Section (Register Protocol No. 97, 20 April 2023—ClinicalTrial.gov registration: NCT01890070), and followed the principles of the Declaration of Helsinki. Once completed, each questionnaire was transmitted to the Google platform, and the final database was downloaded as a Microsoft Excel spreadsheet.

### 2.2. Sample

A total of 501 women completed the questionnaire, with 404 women included in the statistical analysis following data validation. Exclusion criteria comprised (1) pre-pregnancy and gestational diseases, (2) twin pregnancy, (3) vegan or vegetarian diet, (4) smoking habits, (5) personal history of diabetes, (6) alcohol intake during pregnancy, and (7) drug abuse. Maternal fish consumption frequency was categorized as never, 1–2/pw, or ≥3/pw. Gestational outcomes examined included maternal weight gain, mode of delivery (natural or cesarean), and gestational duration. Neonatal outcomes encompassed neonatal weight, height, and the child’s admission to the nursery, neonatology, or neonatal intensive care unit (NICU).

### 2.3. Statistical Analysis

Data for continuous variables are presented as mean and standard deviation, while categorical variables are displayed as absolute and relative frequency tables. Wilcoxon and Kruskal–Wallis tests were conducted to compare means between two or more groups. Pearson’s chi-square and Fisher’s tests were employed for categorical variable tables. Spearman’s non-parametric correlation coefficient was used for numerical variables. The association between fish consumption and the categorical variable of preterm birth was analyzed using multiple logistic regression. All *p*-values below 0.05 were considered statistically significant. Statistical analyses were performed using IBM SPSS for Macintosh (Version 25.0, IBM Corp., Armonk, NY, USA).

## 3. Results

After data validation, 404 women were included in the statistical analysis, as 7 women reported twin pregnancies (including 4 smokers), 15 to be vegetarian (including 11 smokers) and 26 vegan (including 2 smokers), 39 consumed alcohol during pregnancy (including 30 smokers), and with 10 health problems (including 10 smokers). None reported drug abuse. General characteristics and anthropometry of the women are reported in Table 1.

The included women had a mean age of 34 ± 4.8 years (range 23–45). The average pregnancy weight gain was 12.7 ± 4.2 kg (range 0–28). The mean age of patients < 35 years (*n* = 228) was 31.05 ± 2.48, while that of patients ≥ 35 years (*n* = 177) was 38.79 ± 3.27. Regarding weight status, 7.4% were underweight, 75.5% normal weight, 12.6% overweight, and 4.5% obese. Obstetric history showed that 58.0% (235/404) were nulliparous, while 42.0% (169) were multiparous. Details related to neonatal outcomes are presented in Table 2.

Of the 404 women, 74.3% (*n* = 300) had a spontaneous delivery. On average, they gave birth at 39.2 ± 1.39 weeks (range 29–42). The mean birth weight was 3.34 kg ± 457.09 (range 1.969–4.540), and the mean length at birth was 50.42 ± 2.28 cm (range 36–55). The prevalence of low birth weight (<2500 g) was 3.5%, those between 2500 g and 4000 g were 88.9%, and macrosomic newborns (≥4000 g) were 7.6%. Twenty newborns (5%) were admitted to the neonatal intensive care unit (NICU) and five (1.2%) to the special care baby unit (SCBU).

The survey investigated fish consumption during pregnancy (never, 1–2 times a week, or >3 times a week). Out of 404 women, 92.1% (*n* = 372) consumed fish at least once a week, with 67.3% (*n* = 272) doing so 1–2 times a week and 24.8% (*n* = 100) more than 3 times a week, while 8.2% (*n* = 32) never consumed fish. Regarding DHA intake during pregnancy (200 mg/day), 46.0% (*n* = 187) took DHA, while 54.0% (*n* = 217) did not.

Furthermore, 80.7% (*n* = 327) used folic acid supplements, 62.9% (*n* = 255) iron supplements, and 73.1% (*n* = 296) other supplements (vitamin C, vitamin D, myoinositol, magnesium, etc.).

Table 3 presents associations between fish consumption frequencies and gestational/neonatal outcomes.

Table A2 (Appendix B) reports associations between weekly frequency of fish consumption and gestational/neonatal outcomes.

Statistically significant differences were observed in gestational weight gain (GWG) between those not consuming fish and those consuming it ≥3 times a week and between those consuming it 1–2 times a week versus ≥3 times a week (*p*-values = 0.05 and <0.001, respectively). Significant differences in birth weight were observed between those not consuming fish and those consuming it ≥3 times a week (*p*-value = 0.03), as well as in the length of newborns (*p*-value = 0.01).

A statistically significant difference was found between those not taking DHA and those taking it regarding the type of delivery (spontaneous or cesarean), GWG, and weeks of gestation, while birth weight and length at birth were not statistically significant (Table 4). Associations between DHA intake and gestational/neonatal outcomes are reported in Table 4.

Correlations between frequency of fish consumption and gestational/neonatal outcomes were analyzed in the DHA-supplemented and non-DHA-supplemented groups. A negative correlation was observed between fish consumption, both 1–2 t/w and >3 t/w, and DHA supplementation with the GWG (*p* < 0.001 and *p* < 0.001, respectively) and a positive correlation between non-consumption and birth length (*p* < 0.006). No other significant correlations were observed. The results are reported in Table 5.

Spearman’s correlation coefficient was calculated to assess the relationship between fish consumption frequency/pw and gestational/neonatal outcomes in the DHA and non-DHA groups. A negative correlation was observed between increased fish consumption 1–2/pw or ≥3/pw frequency and GWG (−0.03 and −0.08, respectively). DHA assumption negatively influenced GWG (*p* < 0.001) and positively influenced birth length (*p* < 0.019).

Table 6 shows associations between fish consumption or DHA intake and preterm birth (PTB), defined as birth before the 37th week of gestation.

The overall percentage of preterm births in the study population was 5.9% (*n* = 24). No statistically significant associations were found between fish consumption or DHA intake and PTB.

The analysis of the interaction between pre-pregnancy BMI and fish consumption or DHA intake on neonatal outcomes revealed a significant interaction effect in underweight and normal-weight women but not in overweight women. The results are reported in Table 7.

## 4. Discussion

Evidence suggests that fish consumed throughout the first 1000 days, beginning with pregnant and breastfeeding women and extending to infants and children during complementary feeding, can support the optimal growth and development of the child [30]. Fishes are the primary source of *n*-3 [31] and certain minerals and vitamins (selenium, iodine, vitamins A, B12, and D). Adequate intake of *n*-3, particularly DHA, is crucial for optimal neurological development in the fetus and may also offer protection against various adverse perinatal and long-term outcomes [32]. After the first trimester, DHA begins to accumulate rapidly in the brain and continues to accumulate until the age of two [33]. In our sample, 92.1% consume fish at least 1/pw, with an average frequency of 1.7/pw in line with national surveys on Italian food consumption [34]. This frequency in our cohort is higher than that reported in the southern Italy cohort (1.2/pw) [13] but lower than that of the northeastern Italy cohort (2.33/pw) [35]. These data align with all recommendations and guidelines [9,36]. However, fish may contain mercury contamination, which is converted to methylmercury by bacteria and accumulates in the fatty tissue of fish. Exposure to this substance during pregnancy and early childhood is particularly harmful [37] and high levels could be detrimental to the proper development of the fetal nervous system [37,38]. For this reason, all recommendations indicate that pregnant women should not exceed 2–3 servings/pw of low-mercury fish, avoiding large fish [37,38]. An Italian study reported a negative association between maternal consumption of lean fish and shellfish during pregnancy and neonatal head circumference and birth weight [13]. Concurrent exposure to mercury may mask or counterbalance the benefits of fish consumption, especially at high levels of intake. The GMA 2018 also emphasizes that Mediterranean populations tend to have higher mercury (Hg) levels compared to people from Asia, North America, and Europe [39]. In the literature, most studies report a protective effect between fish consumption and birth weight; however, it is significant to highlight that pregnant women in European nations typically ingest approximately 33–43 g of fish daily, a considerably higher amount than the median intake documented in the United States (around 24 g/day) [3] and India (approximately 4 g/day) [40]. EFSA has also stated that pregnant women who consume up to two servings of fish per week are improbable to surpass the provisional tolerable weekly intake (PTWI) established for methylmercury. This assurance holds if they avoid specific fish species such as bluefin tuna or white tuna (not typically utilized in canned tuna production in Europe) and swordfish [41]. In an Italian study where maternal fish intake during pregnancy was moderate (1.69 servings/week), prenatal exposure to Hg was low and did not detect adverse effects on development [35].

Our research revealed a positive correlation between maternal fish intake and higher birth weight. This finding is in line with the most recent review, which shows positive correlations between the consumption of 30 g/day of fatty fish and perinatal outcomes [1], highlighting that women who ate fish more than 1/pw during pregnancy gave birth to infants with higher birth weight than those who rarely ate fish [42]. In our study, infants born to women who consumed fish ≥3 times per week exhibited significantly greater birth length compared to those who did not. However, this finding disagrees with a recent Chinese study in which 10,542 women reported a positive association between total fish consumption and birth weight but no association regarding birth length [16].

We also examined fish consumption with the risk of preterm birth, which in our sample ranges from 32 to 37 weeks, accounting for 5.9% of the total sample, lower than the 6.4% reported by the Italian National Institute of Health in 2019 [43]. We did not find a significant association with fish consumption or DHA supplementation, probably due to the very small sample size (*n* = 24). Exploratory results from the Australian ORIP study [44] suggest that DHA supplementation may need to be increased beyond current prenatal supplement levels for women with low DHA levels. However, the 2022 review concludes that *n*-3 supplementation has not shown a reduced risk of preterm birth compared to a placebo [45].

Unfortunately, pregnant women cannot meet their *n*-3 needs from *n*-3-rich vegetable oils or fish intake alone. Two servings/pw of fish provide only about 100–250 mg/day of *n*-3, including 50–100 mg of DHA [27]. Given the pregnancy dietary target of 650 mg of *n*-3, with 300 mg of DHA [46], exceeding what diet alone provides, the remaining amount can be secured through LCPUFA supplements. Several guidelines recommend during pregnancy an additional daily dose of 200–300 mg of DHA in addition to the 250 mg of EPA and DHA recommended for all adult women of childbearing age [47,48]. Regarding DHA supplementation in our study, there was a statistically significant increase in gestation duration with supplementation (+3 days) compared to the group of women who did not take it, consistent with several studies showing that higher fish oil supplementation increases gestation duration by about 4–4.5 days [49,50,51]. Indeed, *n*-3 was associated with significantly longer gestation duration [52]. It has been proposed that *n*-3 could decrease the activity of prostaglandins F and E, which induce labor, while enhancing the activity of eicosanoids with myometrial relaxation properties, like prostacyclins. This mechanism is thought to prolong gestation [53]. Prolonged gestation positively affects neonatal anthropometry, and, consequently, DHA has been associated with slight improvements in neonatal anthropometry due to its influence on gestation duration. However, our study did not observe such benefits in terms of birth weight and length. This difference may be related to the fact that our sample took a daily dose of 200 mg of DHA during pregnancy, an amount present in most prenatal supplements, much lower than that indicated in the latest Cochrane review of 2018 (500–1000 mg/day) [54].

In our secondary analysis examining the interplay between pre-pregnancy BMI and fish consumption or DHA intake concerning neonatal outcomes, we observed a notable interaction effect in underweight women, whereas no significant interaction was noted among overweight women. Pre-pregnancy BMI is notoriously associated with fetal growth. Underweight BMI has been associated with an increased risk of low birth weight, while pre-pregnancy overweight has been associated with an increased risk of macrosomia and overweight/obese offspring [55]. The observed interaction between pre-pregnancy BMI and fish consumption on birth weight in underweight women aligns with findings from earlier studies, emphasizing the potential benefit of promoting fish consumption among underweight women to mitigate the risk of low-birth-weight infants.

Our study has limitations, as already mentioned, such as not investigating the type of fish consumed during pregnancy. Additionally, although women were selected to reduce risk factors for gestational and neonatal outcomes, other implicated factors were likely not intercepted. Like in most diet and health studies, we used self-reported dietary information during pregnancy, and there is therefore potential for response bias. Finally, besides DHA, it should be acknowledged that many other factors can influence neonatal outcomes, such as maternal diet changes at various stages of pregnancy and the consumption of other important micronutrients like iron and zinc that were not evaluated

Our study has some additional important limitations: First, the dietary data collected are based on self-reported information from participants. This introduces a potential response bias, as memory and subjective perception of dietary habits may not be completely accurate. Although self-reporting is a common methodology in nutritional studies, it is subject to bias that may affect the accuracy of the results. To improve the reliability of the data, future studies could consider objective verification methods, such as the use of nutritional biomarkers (e.g., blood DHA levels) or detailed and continuous monitoring of dietary habits through food diaries or more in-depth interviews. Moreover, there is a lack of heterogeneity in the sample, as a higher percentage of participants had a high level of education (graduate or PhD) compared to the national average. This may limit the generalizability of the results to the general population. Additionally, we did not collect detailed information on the type of fish consumed during pregnancy, which limits the ability to assess the effect of different species, quality, and production method (caught or farmed) on health. It would be useful to further investigate these aspects in future studies to better determine the benefits and risks associated with fish consumption during pregnancy. Finally, we did not collect data on the intake of other micronutrients relevant to maternal and neonatal health, such as iron and zinc [56], which could influence neonatal outcomes. It is crucial to consider that, in addition to DHA, many other dietary and environmental factors can contribute to pregnancy outcomes.

The strength of our study lies in testing women from two regions in central northern Italy, for which specific data on fish consumption or DHA intake were not found. Moreover, to our knowledge, there are still few studies considering both fish consumption and DHA intake during pregnancy.

## 5. Conclusions

Our results support the current advice to pregnant women to include fish and seafood as part of a balanced diet while limiting the intake of species with known high concentrations of environmental pollutants. Given the nutrients provided by fish, its consumption during pregnancy is necessary for fetal and child development [57]. Nevertheless, there remains uncertainty about how to effectively encourage fish consumption without elevating exposure to mercury and other pollutants, which can adversely affect fetal neurological development even at low concentrations. While the use of *n*-3 or fish oil supplements appears to offer a safe means of acquiring the benefits of DHA and EPA without the risk of mercury and toxin exposure, women who opt for *n*-3 fatty acid supplements instead of fish may miss out on several other crucial nutrients found in fish essential for overall health. Future recommendations should involve educating healthcare providers and women on the importance of increasing DHA intake during pregnancy, potentially achieved through a combination of fish consumption and DHA-containing supplements.

## Figures and Tables

**Table 1 nutrients-16-03051-t001:** General maternal characteristics of the sample.

	N Sample = 404
<35 years	31.05 (2.48)
≥35 years	38.79 (3.27)
Pre-pregnancy weight (kg)	60.91 (11.71)
Height (cm)	165.42 (6.34)
Pre-pregnancy BMI (kg/m^2^)	22.0 (3.7)
Underweight	30 [7.4]
Normal weight	305 [75.5]
Pre-obese	51 [12.6]
Obese	18 [4.5]
GWG (kg)	12.7 (4.2)
Nulliparous	235 [58.0]
Multiparous	169 [42.0]
Elementary school	4 [1.0]
Secondary school	123 [30.6]
Degree	213 [52.6]
Ph.D.	64 [15.8]

Values are expressed as mean and IQR (M (IQR)) for continuous variables or as number and percentage (n [%]) for categorical variables. Abbreviations: BMI—body mass index; GWG—gestational weight gain; Ph.D.—Doctor of Philosophy.

**Table 2 nutrients-16-03051-t002:** General child characteristics of the sample.

	N Sample = 404
Males	207 [51.2]
Females	197 [48.8]
Gestation weeks	39.16 (1.39)
Natural childbirth	300 [74.3]
Cesarean delivery	104 [25.7]
Weight < 2500 g	14 [3.5]
Weight 2500–4000 g	359 [88.9]
Weight ≥ 4000 g	31 [7.6]
Height at birth	50.42 (2.28)
Neonatology hospitalization	379 [93.8]
NICU hospitalization	20 [5.0]
SCBU hospitalization	5 [1.2]

Values are expressed as mean and IQR (M (IQR)) for continuous variables or as number and percentage (n [%]) for categorical variables. Abbreviations: NICU—neonatal intensive care unit; SCBU—special care baby unit.

**Table 3 nutrients-16-03051-t003:** Associations between fish consumption and gestational/neonatal outcomes.

Gestational/Neonatal Outcomes	Comparisons	*p*-Value	Std. Error.
**GWG (kg)**	≥3 vs. 0	0.05 *	23.64
	≥3 vs. 1–2	<0.001 **	13.61
	0 vs. 1–2	0.89	21.76
**Gestation weeks**	≥3 vs. 0	0.13	23.46
	≥3 vs. 1–2	0.80	13.51
	0 vs. 1–2	0.07	21.59
**Weight at birth (g)**	≥3 vs. 0	0.04 *	23.71
	≥3 vs. 1–2	0.34	13.66
	0 vs. 1–2	0.09	21.82
**Height at birth (cm)**	≥3 vs. 0	0.01 *	23.41
	≥3 vs. 1–2	0.14	13.48
	0 vs. 1–2	0.08	21.54

Kruskal–Wallis independent samples test. Significance values were adjusted according to Bonferroni’s correction for multiple tests. * *p* < 0.05, ** *p* < 0.005. Abbreviations: GWG—gestational weight gain.

**Table 4 nutrients-16-03051-t004:** Associations between DHA intake and gestational/neonatal outcomes.

Gestational/Neonatal Outcomes	DHA Intake		*p*-Value
	No (*n* = 217)	Yes (*n* = 186)	
Spontaneous delivery	182 (77%)	119 (72%)	<0.001 **
Cesarean delivery	55 (23%)	51 (28%)	
GWG (kg)	12.9 (4.0)	11.9 (3.9)	0.007 *
Gestation weeks	39.0 (1.5)	39.3 (1.2)	0.02 *
Weight at birth (g)	3318.6 (470.1)	3454.8 (441.9)	0.43
Height at birth (cm)	50.3 (2.3)	50.5 (2.3)	0.32

Pearson’s chi-squared test was performed for categorical variables, and Wilcoxon rank sum test was performed for quantitative variables. * *p* < 0.05, ** *p* < 0.005. Abbreviations: GWG—gestational weight gain.

**Table 5 nutrients-16-03051-t005:** Correlations between frequency of fish consumption and gestational/neonatal outcomes in the group of patients taking and not taking DHA.

Gestational/neonatal outcomes	Fish consumption in the group of patients not taking DHA (*n* = 217)		
	Never	1–2 t/w	>3 t/w
	(*n* = 15)	(*n* = 157)	(*n* = 45)
GWG (kg)	−0.076	−0.031	−0.082
	0.298	0.672	0.267
Gestation weeks	0.132	−0.061	0.018
	0.071	0.404	0.807
Weight at birth (g)	0.177	−0.067	0.040
	0.015	0.364	0.584
Height at birth (cm)	0.202	−0.095	−0.026
	0.006 *	0.197	0.723
	Fish consumption in the group of patients taking DHA (*n* = 186)		
	Never	1–2 t/w	>3 t/w
	(*n* = 17)	(*n* = 115)	(*n* = 55)
GWG (kg)	0.024	−0.241	−0.250
	0.720	<0.001 *	<0.001 *
Gestation weeks	0.053	−0.062	−0.035
	0.436	0.364	0.607
Weight at birth (g)	0.013	0.065	−0.080
	0.852	0.337	0.239
Height at birth (cm)	0.004	0.142	0.160
	0.952	0.036	0.019

* *p* < 0.05. Abbreviations: GWG—gestational weight gain; t/w—times per week.

**Table 6 nutrients-16-03051-t006:** Risk of preterm delivery and fish intake or supplementation with DHA.

	Sign.	OR	95% C.I.	
			Inferior	Superior
DHA	0.083	0.449	0.182	1.111
Fish	0.349	0.543	0.151	1.949

Logistic regression model. Abbreviations: DHA—docosahexaenoic acid.

**Table 7 nutrients-16-03051-t007:** Neonatal outcomes in association with fish consumption and DHA supplementation in patients stratified by BMI.

	Gestational Age Birth (days)	*p*-Value	Birth Weight (g)	*p*-Value	Birth Length (cm)	*p*-Value
Pre-pregnancy BMI underweight						
No Fish consumption *n* = 3	36.57 (0.513)	0.004 *	2420.0 (51.962)	0.003 *	45.33 (2.89)	<0.001 **
Fish consumption *n* = 27	39.22 (1.412)		3257.56 (435.23)		40.48 (1.83)	
No DHA intake *n* = 15	39.32 (1.77)	0.21	3305.2 (506.37)	0.14	50.47 (2.23)	0.27
DHA intake *n* = 15	38.59 (1.29)		3042.4 (440.52)		49.47 (2.64)	
Pre-pregnancy BMI normal						
No Fish consumption *n* = 22	39.05 (1.27)	0.55	3153.82 (461.78)	0.06 *	49.32 (2.89)	0.02 *
Fish consumption *n* = 283	39.24 (1.41)		3340.69 (442.16)		50.46 (2.23)	
No DHA intake *n* = 166	39.02 (1.59)	0.006 *	3285.74 (460.87)	0.08	50.18 (2.27)	0.1
DHA intake *n* = 139	39.47 (1.11)		3376.73 (422.59)		50.62 (2.33)	
Pre-pregnancy BMI overweight						
No Fish consumption *n* = 2	38.65 (0.49)	0.73	3050.0 (636.39)	0.28	50.50 (0.71)	0.84
Fish consumption *n* = 49	38.97 (1.27)		3441.20 (490.13)		50.80 (2.06)	
No DHA intake *n* = 24	39.11 (1.27)	0.42	3500.71 (495.66)	0.31	51.02 (2.19)	0.44
DHA intake *n* = 27	38.82 (1.24)		3359.33 (493.23)		50.57 (1.87)	
Pre-pregnancy BMI obese						
No Fish consumption *n* = 5	39.38 (1.15)	0.46	3595.60 (364.06)	0.52	51.50 (1.94)	0.36
Fish consumption *n* = 13	38.87 (1.31)		3444.15 (464.04)		50.38 (2.33)	
No DHA intake *n* = 12	38.41 (0.89)	0.001 *	3426.00 (461.03)	0.42	50.50 (2.39)	0.62
DHA intake *n* = 6	40.22 (0.98)		3606.67 (381.66)		51.08 (2.01)	

For quantitative variables, the Wilcoxon rank sum test was used. * *p* < 0.05, ** *p* < 0.005. Abbreviations: BMI—body mass index; DHA: docosahexaenoic acid.

## Data Availability

The data presented in this study are available on request from the corresponding author due to privacy/ethical restrictions.

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
