# Peer review of "Fish Consumption and DHA Supplementation during Pregnancy: Study of Gestational and Neonatal Outcomes"

_nutrients, 2024, doi:10.3390/nu16183051_

Round 1
Reviewer 1 Report
Comments and Suggestions for Authors
It is an interesting study about fish consumption during pregnancy and its maternal and neonatal effects. It is generally well written, but some major questions and comments were raised during the review process:
1 In the methods section I lack the description of selection process of mothers. How were they selected? Who were contacted? Were they all shortly after delivery when they filled in the online survey? Or mothers with older infants/toddlers were also included?
2 In the exclusion criteria only diabetes is listed. Are mothers with other pre-pregnancy / gestational conditions (e.g.: hypertension, eclampsia, endocrinological disorder, etc.) included? If mothers with pathologies during pregnancy were included, was there a significant influence of fish /DHA supplement intake and the development of these conditions (eclampsia, GDM, etc)? Was use of mind-altering drugs (cannabis or others) not an exclusion criteria?
3 Lines 104-107: it would be also informative to see how many mothers were excluded because of twin pregnancy, vegetarian, vegan diet, alcohol use, smoking (n=?).
4 Based on Table 1 I just wonder, whether your population can represent the general population in Italy, because I feel the number of mothers with higher education (degree, PhD: 68.4%) a bit overrepresented. How many percent of the general population has degree or PhD in Italy? It seems a selection bias for me that might affect your results because those with greater levels of education tend to be more health-conscious with better nutrition. Therefore I think that in the limitations it should be mentioned, that the results of this study can only be generalised with caution.
5 According to the numbers of mothers in Table 1, there must be groups of mothers who both ate fish during pregnancy and took DHA supplements and some mother neither ate fish nor took supplements. So I can see six possible subgroups of mothers. Did you check whether these subgroups (DHA yes, no) in the case of three fish intake groups can be merged (Table 3)? And whether the frequency of fish intake (never, 1-2/week 3/week) influence the results seen in the two DHA supplement groups (Table 4)? Can we see here a reinforcing effect (based on the higher DHA intake) or not?
6 In Table 3 it would be more informative to see whether higher maternal fish intake (3/week vs. no fish intake) resulted in higher or lower gestational weight gain, gestational weeks, and so on (similarly to Table 6), maybe as a supplementary table with all information or in a revised form of this table.
7 In the questionnaire I only can see a question about DHA supplement intake, bot not about the dose. As there are a number of different supplements with very variable doses, I feel this is also a limitation of the study that there is no information about the average daily dose and average daily quantity of DHA. Many supplements only state that it is made of 1000 mg fish oil, but it has a really low DHA content (under 50 mg).
8 As you wrote in the limitations, you have no information about the type of fish consumed by the mothers. Without this information, unfortunately, we have little information on estimating DHA intake during pregnancy, as not all fish types are rich in n-3 LCPUFA. And we also have little information on estimating the potential adverse effects of pollutants.
Author Response
Rome, 27th August 2024
Dear Editor-in-Chief,
First, my coauthors and I sincerely thank you for this cooperation opportunity.
We profoundly thank the Reviewers for the comments and useful suggestions aimed at improving the paper. We thank You for your constructive critique and hope the review process has improved the manuscript. If additional changes are warranted, we will make them.
We hope that this revised version of our manuscript may now be found suitable for publication.
This is a point-by-point list of changes made in the paper:
Reviewer 1
“It is an interesting study about fish consumption during pregnancy and its maternal and neonatal effects. It is generally well written, but some major questions and comments were raised during the review process:
1 In the methods section I lack the description of selection process of mothers. How were they selected? Who were contacted? Were they all shortly after delivery when they filled in the online survey? Or mothers with older infants/toddlers were also included?
The authors thank the Reviewer. Lines 85-91 were revised as follows “The survey was conducted from November 2022 to July 2023, targeting Italian patients who gave birth during that period, belonging to the Division of Gynecology and Ob-stetrics of the Department of Biomedicine and Prevention at Tor Vergata University, and to the Department of Maternal Fetal Medicine of Cristo Re Hospital. Distribution was carried out through an online platform accessible from any internet-connected device, and utilized institutional and private social networks (Twitter, Facebook, and Instagram), the "PATTO in Cucina Magazine" website, and institutional mailing lists, with a snowball method.”
2 In the exclusion criteria only diabetes is listed. Are mothers with other pre-pregnancy / gestational conditions (e.g.: hypertension, eclampsia, endocrinological disorder, etc.) included? If mothers with pathologies during pregnancy were included, was there a significant influence of fish /DHA supplement intake and the development of these conditions (eclampsia, GDM, etc)? Was use of mind-altering drugs (cannabis or others) not an exclusion criteria?
The authors thank the Reviewer. Due to a mistake, not all exclusion criteria were included, so they have been modified as follows: Exclusion criteria comprised (1) pre-pregnancy and gestational diseases, (2) twin pregnancy, (3) vegan or vegetarian diet, (4) smoking habits, (5) personal history of diabetes, (6) alcohol intake during pregnancy, and (7) drugs abuse.
3 Lines 104-107: it would be also informative to see how many mothers were excluded because of twin pregnancy, vegetarian, vegan diet, alcohol use, smoking (n=?).
The authors thank the Reviewer. This information has been included in lines 124-8, as follows: “After data validation, 404 women were included in the statistical analysis, as 7 women reported twin pregnancies (including 4 smokers), 15 to be vegetarian (includ-ing 11 smokers) and 26 vegan (including 2 smokers), 39 consumed alcohol during pregnancy (including 30 smokers), and 10 health problems (including 10 smokers). None reported drug abuse.”
4 Based on Table 1 I just wonder, whether your population can represent the general population in Italy, because I feel the number of mothers with higher education (degree, PhD: 68.4%) a bit overrepresented. How many percent of the general population has degree or PhD in Italy? It seems a selection bias for me that might affect your results because those with greater levels of education tend to be more health-conscious with better nutrition. Therefore I think that in the limitations it should be mentioned, that the results of this study can only be generalised with caution.
The authors thank the Reviewer. From the National Institute of Statistics (ISTAT) we have verified that 65.3% of Italian women have at least a high school diploma, while graduates reach 23.1%. Therefore, as suggested by the Reviewer, the lack of heterogeneity of the sample was included in the limits of the study, as follows: “Moreover, there is a lack of heterogeneity in the sample, as a higher percentage of participants have a high level of education (graduate or PhD) compared to the national average. This may limit the generalizability of the results to the general population.”
5 According to the numbers of mothers in Table 1, there must be groups of mothers who both ate fish during pregnancy and took DHA supplements and some mother neither ate fish nor took supplements. So I can see six possible subgroups of mothers. Did you check whether these subgroups (DHA yes, no) in the case of three fish intake groups can be merged (Table 3)? And whether the frequency of fish intake (never, 1-2/week 3/week) influence the results seen in the two DHA supplement groups (Table 4)? Can we see here a reinforcing effect (based on the higher DHA intake) or not?
The authors thank the Reviewer. A further analysis was performed in order to observe the correlations between frequency of fish consumption and gestational/neonatal outcomes in the group of patients taking (n=186) and not taking DHA (n=217), as reported in the manuscript as follows: “Correlations between frequency of fish consumption and gestational/neonatal outcomes were analyzed in the DHA-supplemented and non-DHA-supplemented groups. A negative correlation was observed between fish consumption, both 1-2 t/w and > 3 t/w, and DHA supplementation with the GWG (p < 0.001 and p < 0.001, re-spectively) and a positive correlation between non-consumption and birth length (p < 0.006). No other significant correlations were observed. The results are reported in Table 5.”
6 In Table 3 it would be more informative to see whether higher maternal fish intake (3/week vs. no fish intake) resulted in higher or lower gestational weight gain, gestational weeks, and so on (similarly to Table 6), maybe as a supplementary table with all information or in a revised form of this table.
The authors thank the Reviewer. A supplementary table (Table A2), regarding the associations between weekly frequency of fish consumption and gestational/neonatal outcomes, has been provided.
7 In the questionnaire I only can see a question about DHA supplement intake, bot not about the dose. As there are a number of different supplements with very variable doses, I feel this is also a limitation of the study that there is no information about the average daily dose and average daily quantity of DHA. Many supplements only state that it is made of 1000 mg fish oil, but it has a really low DHA content (under 50 mg).
The authors thank and agree with the Reviewer. However, supplements available in Italy all have a dose of 200 mg of DHA, therefore this information has been reported in the manuscript, as follows: “Regarding the doses of DHA, no questions were asked, as the supplements available in Italy have a dose of 200 mg.”.
8 As you wrote in the limitations, you have no information about the type of fish consumed by the mothers. Without this information, unfortunately, we have little information on estimating DHA intake during pregnancy, as not all fish types are rich in n-3 LCPUFA. And we also have little information on estimating the potential adverse effects of pollutants.”
The authors thank and agree with the Reviewer. The limitation of the study was therefore further highlighted and a further study was encouraged, which considers the typology, quality and type of production (caught/farmed) of the fish consumed. The limitations were revised as follows: “Additionally, we did not collect detailed information on the type of fish consumed during pregnancy, which limits the ability to assess the effect of different species, quality, and production method (caught or farmed) on health. It would be useful to further investigate these aspects in future studies to better determine the benefits and risks associated with fish consump-tion during pregnancy.”
We confirm that this manuscript describes original work and is not under consideration by any other journal. All authors approved the manuscript and this submission.
We declare no conflict of interest.
We look forward to hearing from you at your earliest convenience.
Yours sincerely,
Rossella Cianci
Reviewer 2 Report
Comments and Suggestions for Authors
The article ‘Fish consumption and DHA supplementation during pregnancy: study of gestational and neonatal outcomes’ addresses important issues related to pregnant women's diet and its impact on fetal development. The article shows several strengths, but at the same time has some limitations that are worth considering before accepting it.
The strengths of the article include the timeliness and relevance of the topic addressed, which is important from a public health perspective. The study was conducted reliably, using appropriate statistical methods, which adds to the reliability of the results. The authors analysed a wide range of variables, which allows for a comprehensive assessment of the impact of diet on pregnancy and neonatal outcomes.
However, the article also has some limitations. Firstly, there is a lack of detailed information on the type of fish consumed, which is important given the different levels of contaminants and nutritional values in different fish species. Secondly, the article focuses mainly on DHA and fish consumption, leaving out other key nutrients that may affect health outcomes. Another limitation is the reliance on self-reported data, which may lead to errors due to inaccuracy or participant bias.
To improve the quality of the article and increase its scientific value, the authors should consider adding information on the species of fish consumed by the female participants in the study. This will allow a more precise analysis of the potential benefits and risks associated with their consumption. It would also be important to include other micronutrients that may affect maternal and child health, allowing for a better understanding of the relationships studied. Additionally, it would be useful to consider the limitations of self-reporting data and possibly introduce verification methods to minimise the risk of errors.
These recommendations can contribute to the reliability and credibility of the article, which is crucial for its acceptance for publication.
Author Response
Rome, 27th August 2024
Dear Editor-in-Chief,
First, my coauthors and I sincerely thank you for this cooperation opportunity.
We profoundly thank the Reviewers for the comments and useful suggestions aimed at improving the paper. We thank You for your constructive critique and hope the review process has improved the manuscript. If additional changes are warranted, we will make them.
We hope that this revised version of our manuscript may now be found suitable for publication.
This is a point-by-point list of changes made in the paper:
Reviewer 2
“The article ‘Fish consumption and DHA supplementation during pregnancy: study of gestational and neonatal outcomes’ addresses important issues related to pregnant women's diet and its impact on fetal development. The article shows several strengths, but at the same time has some limitations that are worth considering before accepting it. The strengths of the article include the timeliness and relevance of the topic addressed, which is important from a public health perspective. The study was conducted reliably, using appropriate statistical methods, which adds to the reliability of the results. The authors analysed a wide range of variables, which allows for a comprehensive assessment of the impact of diet on pregnancy and neonatal outcomes.
However, the article also has some limitations. Firstly, there is a lack of detailed information on the type of fish consumed, which is important given the different levels of contaminants and nutritional values in different fish species. Secondly, the article focuses mainly on DHA and fish consumption, leaving out other key nutrients that may affect health outcomes. Another limitation is the reliance on self-reported data, which may lead to errors due to inaccuracy or participant bias.
To improve the quality of the article and increase its scientific value, the authors should consider adding information on the species of fish consumed by the female participants in the study. This will allow a more precise analysis of the potential benefits and risks associated with their consumption.
The authors thank the Reviewer. However, this information was not requested in the questionnaire, thus it was further highlighted in the limitations of the study, as follows: “Additionally, we did not collect detailed information on the type of fish consumed during pregnancy, which limits the ability to assess the effect of different species, quality, and production method (caught or farmed) on health. It would be useful to further investigate these aspects in future studies to better determine the benefits and risks associated with fish consumption during pregnancy.”
It would also be important to include other micronutrients that may affect maternal and child health, allowing for a better understanding of the relationships studied.
The authors agree with the Reviewer and thank for the valuable suggestion. Not having specific data or that can be extrapolated regarding the other micronutrients, the lack was highlighted more within the limits of the study, as follows: “Finally, we did not collect data on the intake of other micro-nutrients relevant to maternal and neonatal health, such as iron and zinc [57], which could influence neonatal outcomes. It is crucial to consider that, in addition to DHA, many other dietary and environmental factors can contribute to pregnancy outcomes.”
Additionally, it would be useful to consider the limitations of self-reporting data and possibly introduce verification methods to minimise the risk of errors.
These recommendations can contribute to the reliability and credibility of the article, which is crucial for its acceptance for publication.
The authors thank the Reviewer. Self-reporting of data was further highlighted within the limitations of the study, as follows: “Our study has some important limitations. First, the dietary data collected is based on self-reported information from participants. This introduces a potential response bias, as memory and subjective perception of dietary habits may not be completely accurate. Although self-reporting is a common methodology in nutritional studies, it is subject to bias that may affect the accuracy of the results. To improve the reliability of the data, future studies could consider objective verification methods, such as the use of nutritional biomarkers (e.g., blood DHA levels) or detailed and continuous monitoring of dietary habits through food diaries or more in-depth interviews.”
We confirm that this manuscript describes original work and is not under consideration by any other journal. All authors approved the manuscript and this submission.
We declare no conflict of interest.
We look forward to hearing from you at your earliest convenience.
Yours sincerely,
Rossella Cianci
Round 2
Reviewer 1 Report
Comments and Suggestions for Authors
I accept the answers and corrections made by the authors. The manuscript improved a lot and the additional tables support the findings. I only have some minor comments:
1 Table 1: the unit of BMI is kg/m2, so delete ‘c’ and put 2 in superscript.
2 Line 143: unit is missing for weight gain, please add.
3 Table 2: in rows 6-12 (weight <2500g-SCBU hospitalization) data don’t look good (e.g. weight <2500g cannot be 3335.35, it seems rather average birth weight to me, as you also write in the text line 153). It seems that a row is missing from the left part. Please revise this table.
Author Response
​​​​​​​​​​​
Rome, 30th August 2024
Dear Editor in Chief,
First, my coauthors and I would like to thank You sincerely for this opportunity of cooperation.
We profoundly thank the Reviewers for the comments and useful suggestions aimed at improving the paper. We thank You for your constructive critique and we hope the review process has led to an improved manuscript. If additional changes are warranted, we will make them.
We hope that this revised version of our manuscript may now be found suitable for publication.
This is a point-by-point list of changes made in the paper:
Reviewer 1
“I accept the answers and corrections made by the authors. The manuscript improved a lot and the additional tables support the findings. I only have some minor comments:
1 Table 1: the unit of BMI is kg/m2, so delete ‘c’ and put 2 in superscript.
The authors thank the Reviewer for the carefulobservation regarding the unit of BMI in Table 1. The error have been corrected by removing the 'c' and placing the '2' in superscript.
2 Line 143: unit is missing for weight gain, please add.
The authors thank the Reviewer for pointing out the missing unit for weight gain in line 143. The appropriateunit have been added.
3 Table 2: in rows 6-12 (weight <2500g-SCBU hospitalization) data don’t look good (e.g. weight <2500g cannot be 3335.35, it seems rather average birth weight to me, as you also write in the text line 153). It seems that a row is missing from the left part. Please revise this table.
The authors thank the Reviewer. It appears that there wasa typo in Table 2, so authors have corrected the data as suggested. The table has now been revised accordingly.
We confirm that this manuscript describes original work and is not under consideration by any other journal. All authors approved the manuscript and this submission.
We declare no conflict of interest.
We look forward to hearing from you at your earliest convenience.
Yours sincerely,
Rossella Cianci
Reviewer 2 Report
Comments and Suggestions for Authors
Dear Authors,
The manuscript has been substantially revised.
Author Response
​​​​​​​​​​​Rome, 30th August 2024
Dear Editor in Chief,
First, my coauthors and I would like to thank You sincerely for this opportunity of cooperation.
We profoundly thank the Reviewers for the comments and useful suggestions aimed at improving the paper. We thank You for your constructive critique and we hope the review process has led to an improved manuscript. If additional changes are warranted, we will make them.
We hope that this revised version of our manuscript may now be found suitable for publication.
This is a point-by-point list of changes made in the paper:
Reviewer 2
“Dear Authors,
The manuscript has been substantially revised.”
The authors thank the Reviewer.
We confirm that this manuscript describes original work and is not under consideration by any other journal. All authors approved the manuscript and this submission.
We declare no conflict of interest.
We look forward to hearing from you at your earliest convenience.
Yours sincerely,
Rossella Cianci